# A Comparison Study on Toughening Vinyl Ester Resins Using Different Nanocarbon Materials

**DOI:** 10.3390/polym15234536

**Published:** 2023-11-25

**Authors:** Ruirui Yang, Yating Wang, Xiaolu Ran, Wanshuang Liu

**Affiliations:** 1Luoyang Ship Material Research Institute, 169 South Binhe Road, Luoyang 471023, China; cute725@126.com (R.Y.); wangyt1232023@163.com (Y.W.); 2Shanghai High Performance Fibers and Composites Center (Province-Ministry Joint), College of Textiles, Donghua University, 2999 North Renmin Road, Shanghai 201620, China; rxl180402115@163.com

**Keywords:** vinyl ester resins, toughening effects, nanocarbon materials, polymer composites

## Abstract

This study aims to comprehensively compare and evaluate the toughening effects of different nanocarbon materials on vinyl ester resins. Four typical nanocarbon materials, including graphene, graphene oxide (GO), single-walled carbon nanotubes (SWCNTs), and multi-walled carbon nanotubes (MWCNTs), were used as reinforcing fillers for vinyl ester resins. These four nanocarbon materials were dispersed in the vinyl ester resin matrix by the combination of high-speed stirring and probe sonication, and their dispersion states were observed with optical microscopy. The effects of incorporating different nanocarbon materials on the viscosities, thermal properties, tensile properties, and fracture toughness of the resulting modified vinyl ester resins were systematically investigated. The results indicate that the four nanocarbon materials show enhanced toughening effects on the vinyl ester resin in the sequence of SWCNTs, GO, MWCNTs, and graphene. Compared with the control resin, the modified vinyl ester resin containing 0.2 wt% graphene shows 45% and 54% enhancements in the critical stress intensity factor (*K*_IC_) and critical strain energy release rate (*G*_IC_), respectively. The incorporation of the four nanocarbon materials has almost no effect on the glass transition temperatures of the resulting modified vinyl ester resins. This study provides valuable insights into the selection of nanocarbon additives for enhancing the toughness of vinyl ester resins.

## 1. Introduction

Vinyl ester resins, a versatile class of thermosetting polymers, have been widely used in protective coatings, chemical processing parts, construction materials, and polymer matrices for fiber-reinforced composites owing to their exceptional combination of mechanical strength, corrosion resistance, and ease of processing [1,2,3]. However, vinyl ester resins are inherently brittle and have poor resistance to impact failure, which greatly limits their industrial applications. Adding a secondary phase (rigid or flexible) is a widely recognized approach for enhancing the fracture toughness of vinyl ester resins. It has been reported that vinyl ester resins can be effectively toughened by thermoplastic elastomers, rubber particles, or rigid nanoparticles [4,5,6,7,8].

Recently, nanocarbon materials, such as carbon nanotubes (CNTs) [9,10], carbon nanofibers (CNFs) [11,12], graphene nanoplatelets [13,14,15], and graphene oxides (GO) [16,17], have been regarded as a class of promising reinforcing fillers for thermosetting polymers due to their large surface area, exceptional mechanical properties, light weight, and low filler loading. In previous reports, various pristine or functional nanocarbon materials have been mainly used to toughen thermosetting epoxy resins. Only a few reports are related to the toughening of vinyl ester resins. For instance, Stein and coworkers prepared dodecylamine functionalized GO (mGO) and used it to toughen vinyl ester resins [7]. When only 0.02 wt% mGO was added, the critical stress intensity factor (*K*_IC_) and the critical strain energy release rate (*G*_IC_) of the resulting modified vinyl ester resin increased by approximately 18 and 42%, respectively. Owing to the low filler loading, the flexural strength and modulus of the modified vinyl ester resin remained nearly unchanged. Nevertheless, toughening vinyl ester resins by CNTs, graphene, and GO has scarcely been reported, to the best of our knowledge.

It has been reported that the types, morphologies, dispersion methods, and synthesis technologies of incorporated nanocarbon materials are closely related to the properties of the resulting polymer composites [18,19]. Schulte and coworkers performed a comparative study on the mechanical performances of epoxy resins modified by different carbon nanotubes [20]. They evaluated single-walled carbon nanotubes (SWCNTs), double-walled carbon nanotubes (DWCNTs), and multi-walled carbon nanotubes (MWCNTs). These three CNTs show increased toughening effects on epoxy resins in the sequence of SWCNTs, MWCNTs, and DWCNTs at 0.3 wt% filler loading. Domun and coworkers reviewed the effects of the incorporation of various nanocarbon materials on the fracture toughness, strength, and stiffness of epoxy resins [19]. The types of nanocarbon fillers and interface interactions between nanocarbon fillers and polymer matrices markedly affect the mechanical properties of the resulting polymer composites.

To date, how the addition of different nanocarbon materials affects the mechanical properties of vinyl ester resins is still unexplored in the literature. Therefore, a better understanding of the toughening effects of various nanocarbon materials on vinyl ester resins through a comparative study is necessary. Graphene, GO, SWCNTs, and MWCNTs, as four typical nanocarbon materials, have quite different morphologies, specific surface areas, and surface functional groups, which greatly affects their dispersibility and toughening effects on the polymer matrix. In this study, graphene, GO, SWCNTs, and MWCNTs were used to reinforce vinyl ester resins. Different dispersion methods, including high-speed stirring by a homogenizer, probe sonication, three-roll milling, and planetary centrifugal mixing, were used to disperse the nanocarbon fillers in the vinyl ester resin matrix. In addition to fracture toughness, the rheological, thermal, and tensile properties of the vinyl ester resins modified by different nanocarbon fillers were also investigated.

## 2. Materials and Methods

### 2.1. Materials

SWCNTs (diameter: 1–2 nm, length: 10–30 μm), MWCNTs (diameter: 5–15 nm, length: 10–50 μm), graphene nanoplatelets (diameter: 5–10 μm, thickness: 1–5 nm), and GO (diameter: 0.5–5 μm, thickness: 0.8–1.2 nm) were purchased from Nanjing XFNANO Materials Tech Co., Ltd., Nanjing, China. The dimension information of the four nanocarbon materials is according to the supplier’s specifications. Vinyl ester resin (MFE-2) and curing agents (M-50 and NL-49P) were supplied by Sino Polymer Co. Ltd., Shanghai, China.

### 2.2. Preparation of Nanocarbon/Vinyl Ester Resin Composites

A calculated amount of nanocarbon fillers (0.2 wt%) was added to the liquid vinyl ester resin system. The weight ratio of MFE-2/M50/NL49P was 97.7/1.5/1.0. Five methods were used to disperse the nanocarbon fillers in the vinyl ester resin matrix. The first method is to disperse the mixture with a homogenizer with a stirring speed of 3000 rpm. The second method is to disperse the mixture with a probe ultrasonic machine (SCIENTZ-IID, 1 cm^2^ Ti horn with a pulse: 2 s on and 2 s off) at 75% amplitude for 30 min. The third method is to disperse the mixture with a three-roll milling machine (ZYTR-80E) five times. The fourth method is to disperse the mixture by planetary centrifugal mixing (ZYMB-4000VS) for 2 min. The fifth method is to disperse the mixture through a combination of the first and second methods. After degassing, the mixture was poured into metal molds with specific dimensions for various tests. All vinyl ester resin samples were cured at 25 °C for 24 h and 80 °C for 12 h. The control vinyl ester resin was cured using the same conditions.

### 2.3. Characterization

The viscosities of all the vinyl ester resin samples before curing were measured using a Brookfield DV2T plate rheometer at 25 °C with a rotation speed of 5.0 rpm. The dispersion states of various nanocarbon materials in the vinyl ester resin matrix were observed using a Puda FM-400C fluorescence microscope. The modified vinyl ester resin mixture was coated on a glass slide for observation. The morphologies of fracture surfaces from tensile tests were observed on a Hitachi S4800 scanning electron microscope (SEM). The samples were coated with gold for 90 s before SEM observation. Dynamic mechanical analysis (DMA) was performed on a TA Instruments DMA Q800 with a heating rate of 3 °C min^−1^ and a frequency of 1 Hz. Double cantilever mode was used for the DMA tests and the specimen dimensions were 60 × 15.0 × 2.0 mm^3^. Flexural properties were tested on a Wance ETM104B-EX electronic universal testing machine following the ASTM D790 standard [21]. The load cell is 2000 N and the crosshead speed is 1 mm min^−1^. Fracture toughness was assessed through a single-edge-notch bending configuration in accordance with the ASTM D5045 standard [22]. To initiate a precrack, a chilled blade was carefully tapped at the base of a saw slot positioned in the center of each specimen. The reported values for both tensile and fracture toughness tests represent the averages obtained from six valid test specimens.

## 3. Results

### 3.1. Dispersion of Nanocarbon Fillers

It is known that the dispersion states of nanocarbon fillers in the polymer matrix are closely related to their reinforcing effects. In this work, four types of dispersion equipment were used to disperse nanofillers in the vinyl ester resin matrix, as mentioned in the experimental section (Figure 1). To optimize the dispersion methods, MWCNTs were representatively chosen to be dispersed in the vinyl ester resin matrix by homogenizer, probe sonication, three-roll milling, and planetary centrifugal mixing. The dispersion states of MWCNTs with different dispersion methods were observed using an optical microscope, as shown in Figure 2. It is clear that the MWCNTs tend to agglomerate in the vinyl ester resin matrix after the three-roll milling treatment. This might be because the gap between the rolls is at the micrometer level, which is not suitable for nanoscale fillers. In contrast, the dispersion states of MWCNTs were much better after dispersing with the other three methods. Based on the above results, all the nanocarbon fillers were dispersed by the combination of high-speed stirring and probe sonication in the following study. Figure 3 shows the optical microscopy observations of various nanocarbon fillers in a vinyl ester resin matrix with the combined dispersing methods. For the mixtures containing graphene, GO and MWCNTs, scattered black spots, and no large aggregations can be observed (Figure 3a–c). The SWCNTs show a fibrous morphology with different sizes in the mixture (Figure 3d).

### 3.2. Viscosity

The viscosities of vinyl ester resin mixtures containing different nanocarbon materials (0.2 wt%) were measured with a Brookfield plate rheometer, and the results are shown in Figure 4. The control vinyl ester resin has a very low viscosity value of 155 mPa.s. The addition of four nanocarbon fillers shows quite different impacts on the viscosities of the resulting vinyl ester resin mixtures, which increase in the sequence of MWCNTs (345 mPa.s), graphene (483 mPa.s), GO (725 mPa.s), and SWCNTs (2385 mPa.s). The highest viscosity of the SWCNTs/vinyl ester resin mixture is due to the high specific surface area of SWCNTs. Compared with graphene, the addition of GO leads to a greater increase in viscosity. This might be because the polar groups (-OH and -COOH) on the surface of GO could increase the interfacial interactions between GO and the vinyl ester resin matrix through hydrogen bonding.

### 3.3. Mechanical Properties

To investigate the effects of the four nanocarbon materials on the mechanical properties of the vinyl ester resin, tensile and fracture toughness measurements were conducted. Figure 5a shows representative tensile stress–strain curves of the control vinyl ester resin and four composites. Their tensile strength and modulus are summarized in Figure 5b,c. The tensile strength and modulus of the control vinyl ester resin are 57.7 and 1043 MPa, respectively. Compared with the control vinyl ester resin, the addition of 0.2 wt% graphene, GO, MWCNTs, and SWCNTs gives 47, 24, 25, and 6% increases in tensile strength and 23, 17, 22, and 13% increases in tensile modulus, respectively. The fracture toughness results of the control vinyl ester resin and four composites are shown in Figure 6. The critical stress intensity factor (*K*_IC_) and critical strain energy release rate (*G*_IC_) of the control vinyl ester resin are 0.80 MPa.m^1/2^ and 546 J m^−2^, respectively. Compared with the control vinyl ester resin, the composites containing 0.2 wt% graphene, GO, MWCNTs, and SWCNTs exhibit 45, 22, 13, and 14% increases in *K*_IC_ and 54, 26, 37, and 18% increases in *G*_IC_, respectively. The above results indicate that these four nanocarbon materials show increased reinforcing (or toughening) effects in the sequence of SWCNTs, GO, MWCNTs, and graphene. Similarly, graphene also shows superiority over carbon nanotubes for toughening epoxy resins due to its high specific surface area and two-dimensional geometry [23]. SWCNTs and GO show inferior reinforcing effects to vinyl ester resins, which might be because of their poor dispersibility, as shown in Figure 3.

The fracture surfaces of the control vinyl ester resin and four composites after tensile tests were observed using SEM (Figure 7). The control vinyl ester resin presents a smooth fracture surface with some river-like patterns (Figure 7a), indicating a brittle failure mode without any ductility. In contrast, four composites show rough fracture surfaces with an off-plane and tortuous cracks (Figure 7b–e). The incorporation of nanocarbon materials would cause crack deflection and result in multiplane fracture surfaces. The generation of new cracks and fracture surfaces could dissipate more energy during the failure process. It should be noted that numerous fine cracks and small resin blocks can be observed on the fracture surface of the composite containing SWCNTs (Figure 7e). This might be because of the smaller size of SWCNTs and the stress concentration induced by the agglomeration of SWCNTs. The precrack propagation regions of the control vinyl ester resin and four composites after fracture toughness tests were observed with optical microscopy, as shown in Figure 8. The control vinyl ester resin shows a smooth fracture surface with a small number of straight cracks (Figure 8a), suggesting that the crack propagated steadily in an uninterrupted manner. In contrast, the composites containing graphene, GO, and MWCNTs present relatively rough fracture surfaces with more cracks (Figure 8b–d), which is consistent with their increased fracture toughness. It should be noted that the cracks on the fracture surfaces of the composite containing SWCNTs became less noticeable (Figure 8e). This reflects that the addition of SWCNTs cannot effectively toughen the vinyl ester resin matrix. The toughening mechanisms of rigid nanocarbon fillers have been widely reported, and mainly include crack deflection, crack pinning, crack bridging, and plastic yielding of the matrix [24,25,26,27].

### 3.4. Dynamic Mechanical Analysis

Thermo-mechanical properties are important performance indices for polymer materials because they usually determine the ceiling temperature for applications. DMA tests were conducted to investigate the influence of adding four nanocarbon fillers on the thermo-mechanical properties of vinyl ester resin. The curves of storage modulus, loss modulus, and tan δ versus temperature are presented in Figure 9, and the related data are summarized in Table 1. The storage modulus is often used to describe the stiffness or elasticity of a material. Compared with the control vinyl ester resin, the composites containing graphene, GO, and MWCNTs display increased storage moduli in the glassy region (40 °C), as shown in Figure 9a and Table 1. In general, the incorporation of rigid nanocarbon fillers can increase the rigidity of the polymer matrices [19]. The composites containing graphene and MWCNTs show a noticeably higher storage modulus than the other samples. However, the composite containing SWCNTs shows a decreased storage modulus (Table 1). This might be because the agglomeration of SWCNTs would be unfavorable for the load transfer between the vinyl ester resin and SWCNTs.

The loss modulus is an important rheological parameter that reflects the energy dissipation in a material when subjected to cyclic stress. As shown in Figure 9b and Table 1, the composites containing graphene and MWCNTs show increased maximum values of loss modulus compared with those of the control vinyl ester resin. This indicates that the incorporation of graphene and MWCNTs might cause additional internal friction between the polymer chains and fillers. In contrast, the composite containing SWCNTs shows decreased maximum loss modulus values. This might be because the addition of SWCNTs with thin diameters could restrict the chain movement of vinyl ester resin. The incorporation of GO has almost no effect on the maximum loss modulus of the vinyl ester resin matrix.

Tan δ is the ratio of the loss modulus to the storage modulus, and its maximum value at a specific temperature is often taken as the glass transition temperature (*T*_g_). As shown in Figure 9c and Table 1, the composites containing graphene, GO, and MWCNTs show comparable *T*_g_ values (120–122 °C) in comparison to the control vinyl ester resin (121 °C). The composite containing SWCNTs shows a slightly increased *T*_g_ value (125 °C). This further indicates that the incorporation of SWCNTs could restrict the chain movement of vinyl ester resin to some extent, which is consistent with the loss modulus results for the composite containing SWCNTs. The above results suggest that toughening vinyl ester resin with these four nanocarbon materials will not sacrifice the heat resistance of the polymer matrix.

## 4. Conclusions

In summary, four typical nanocarbon materials, including graphene, GO, MWCNTs, and SWCNTs, were used to modify vinyl ester resin, and a comparative study was performed. The main conclusions are as follows. First, high-speed stirring and probe-type sonication are more suitable for dispersing the nanocarbon filler into the vinyl ester resin matrix among the four adopted dispersion methods. Second, the addition of nanocarbon fillers increases the viscosities of the resulting vinyl ester resin mixtures in the sequence of MWCNTs, graphene, GO, and SWCNTs. Third, graphene shows the best reinforcing effect on vinyl ester resin among the four nanocarbon fillers. The modified vinyl ester resin containing 0.2 wt% graphene shows 45 and 54% increases in *K*_IC_ and *G*_IC_, respectively. The addition of SWCNTs has the least impact on the mechanical properties of the vinyl ester resin matrix. Fourth, microscopic observations indicate that the modified vinyl ester resin containing 0.2 wt% graphene presents clear crack deflections and increased crack quantity on its fracture surface, which is consistent with the superior reinforcing effect of graphene. Finally, the results of the DMA measurements indicate that the addition of four nanocarbon fillers has no significant impact on the *T*_g_ values of the resulting modified vinyl ester resins. Considering that reports on toughening vinyl ester resins using nanocarbon materials are very scarce, we believe that the comparative study in this work will contribute to a better understanding of the toughening effects of nanocarbon fillers on vinyl ester resins. In future work, the effects of the surface functionalization of nanocarbon materials on interfacial interactions (related to load transfer) and mechanical properties of modified vinyl ester resins will be systematically investigated.

## Figures and Tables

**Figure 1 polymers-15-04536-f001:**
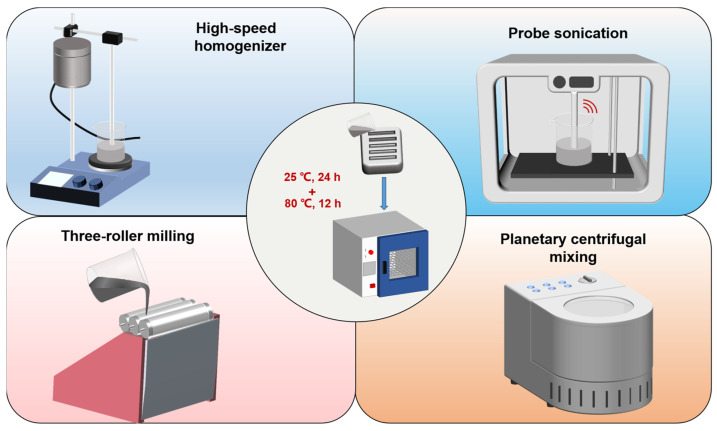
Schematic diagrams of the preparation of nanocarbon-modified vinyl ester resin using different dispersion methods.

**Figure 2 polymers-15-04536-f002:**
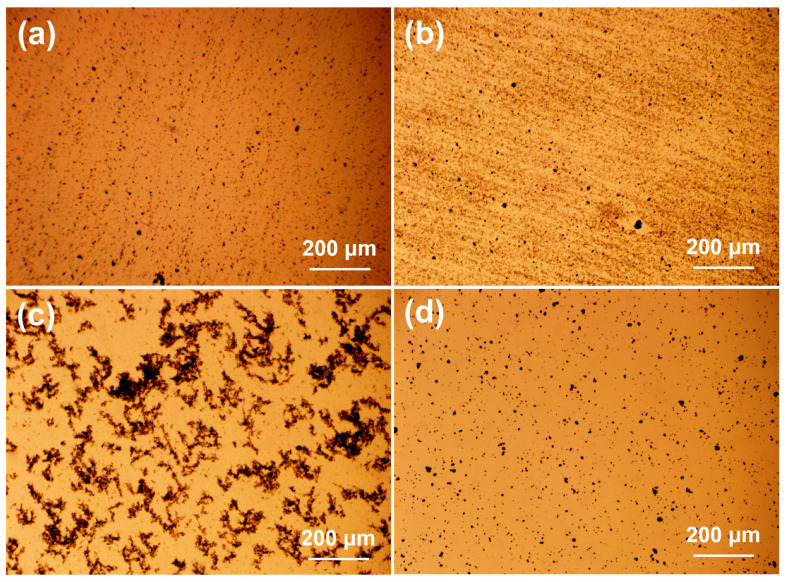
Optical micrographs of MWCNTs (0.2 wt%) in vinyl ester resin after dispersing with homogenizer (**a**), probe sonication (**b**), three-roll milling (**c**), and planetary centrifugal mixing (**d**). The scale bars in the micrographs are 200 μm.

**Figure 3 polymers-15-04536-f003:**
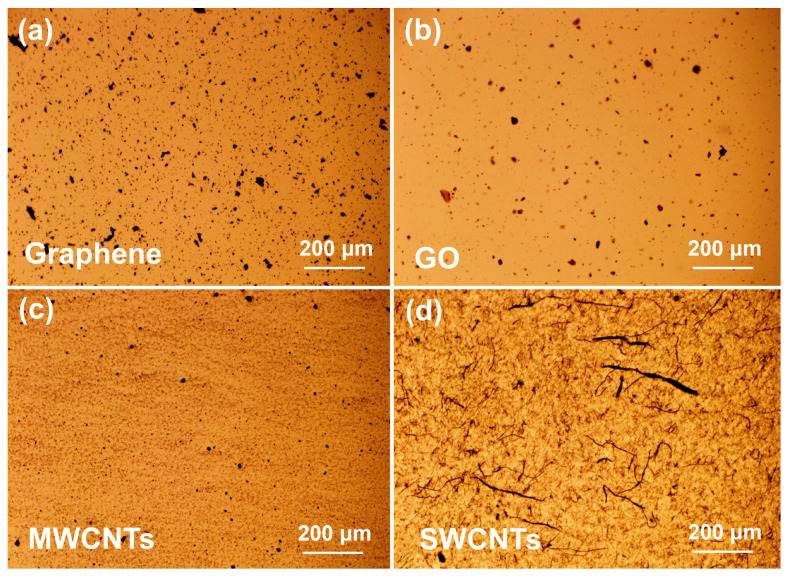
Optical micrographs of graphene (**a**), GO (**b**), SWCNTs (**c**), and MWCNTs (**d**) in vinyl ester resin after dispersing with homogenizer and probe sonication. The filler content is 0.2 wt% and the scale bars in the micrographs are 200 μm.

**Figure 4 polymers-15-04536-f004:**
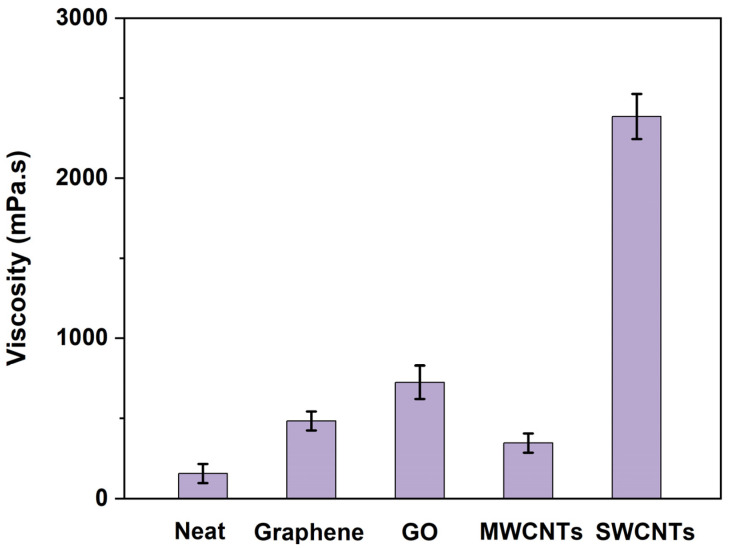
Viscosities of control vinyl ester resin and four nanocarbon/vinyl ester resin mixtures at 25 °C.

**Figure 5 polymers-15-04536-f005:**
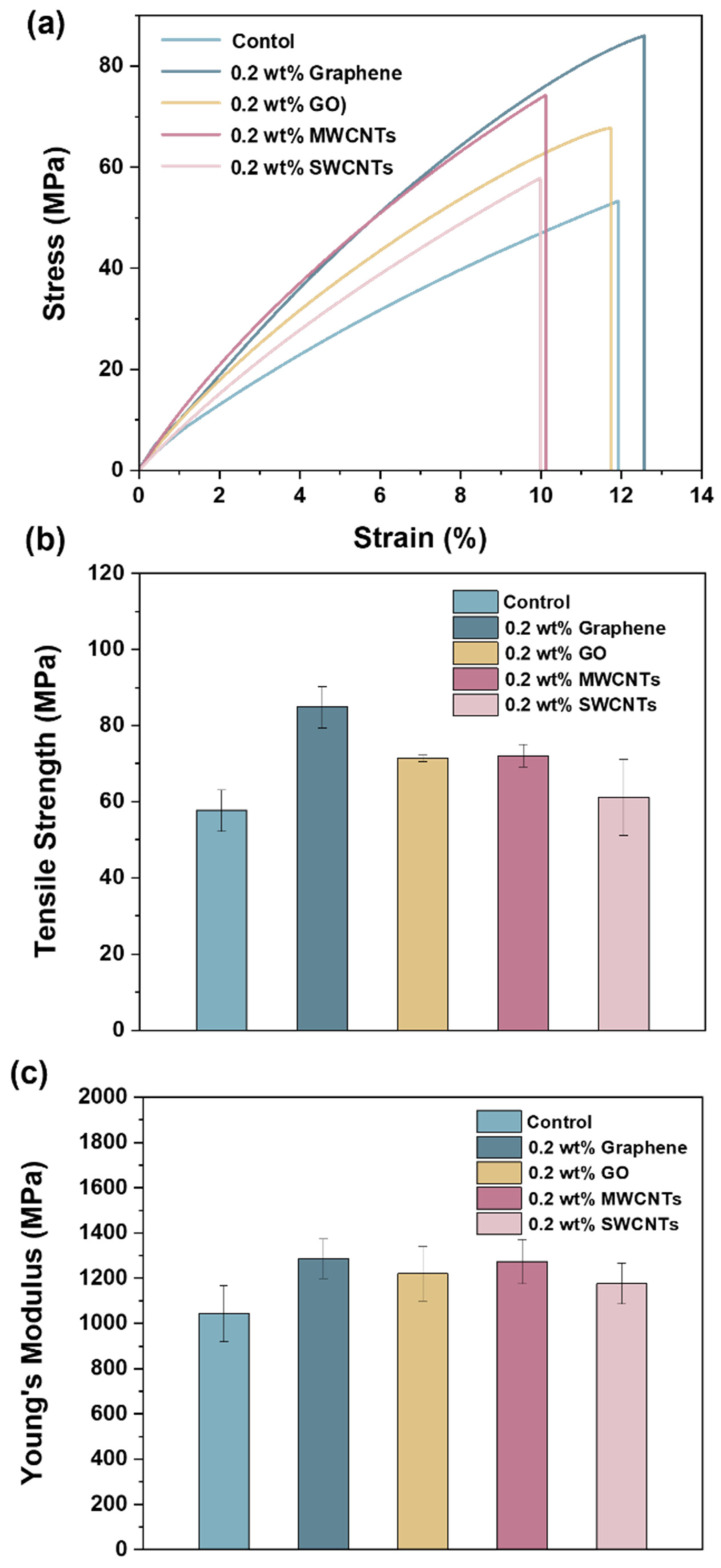
Representative tensile stress–strain curves (**a**), tensile strength (**b**), and tensile modulus (**c**) of control vinyl ester resin and four vinyl ester resin/nanocarbon composites.

**Figure 6 polymers-15-04536-f006:**
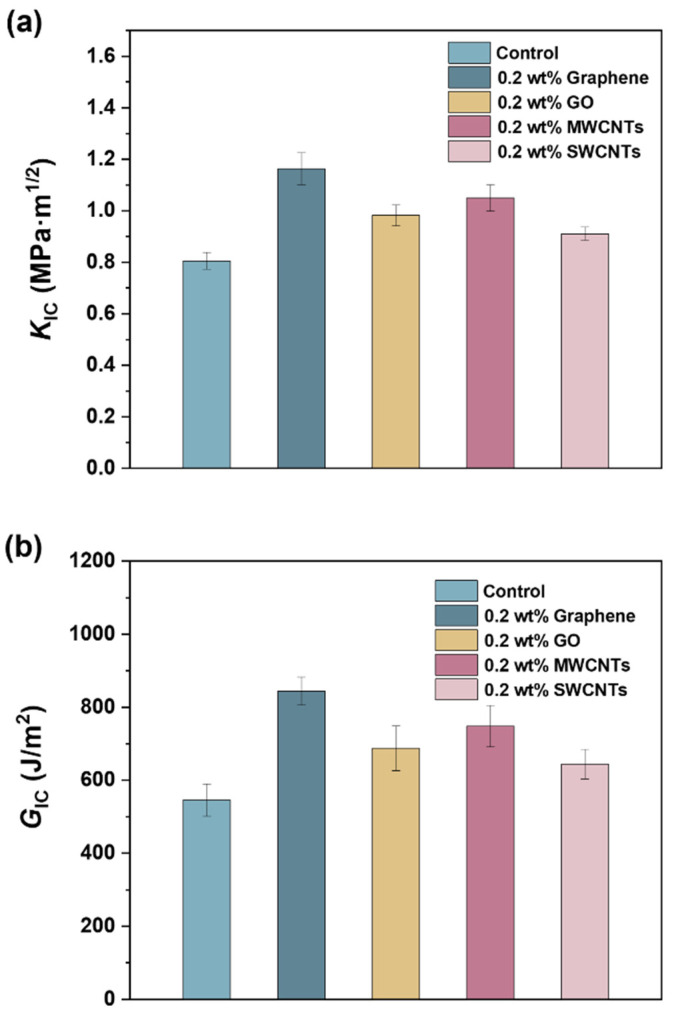
Critical stress intensity factor (**a**) and critical strain energy release rate (**b**) of control vinyl ester resin and four vinyl ester resin/nanocarbon composites.

**Figure 7 polymers-15-04536-f007:**
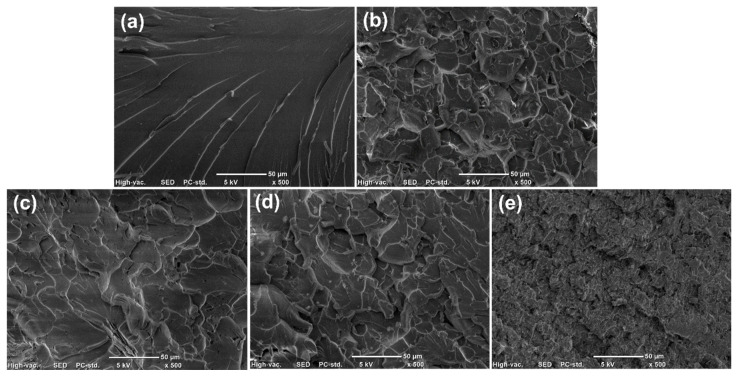
SEM micrographs of the control vinyl ester resin (**a**) and four composites containing graphene (**b**), GO (**c**), MWCNTs (**d**), and SWCNTs (**e**). The scale bars in the micrographs are 50 μm.

**Figure 8 polymers-15-04536-f008:**
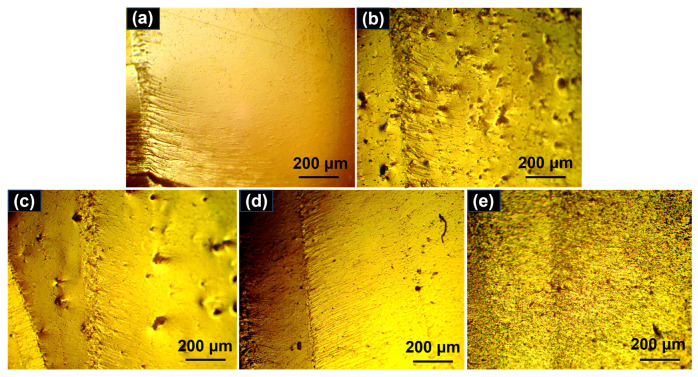
Optical microscopy micrographs of the control vinyl ester resin (**a**) and four composites containing graphene (**b**), GO (**c**), MWCNTs (**d**), and SWCNTs (**e**). The scale bars in the micrographs are 200 μm.

**Figure 9 polymers-15-04536-f009:**
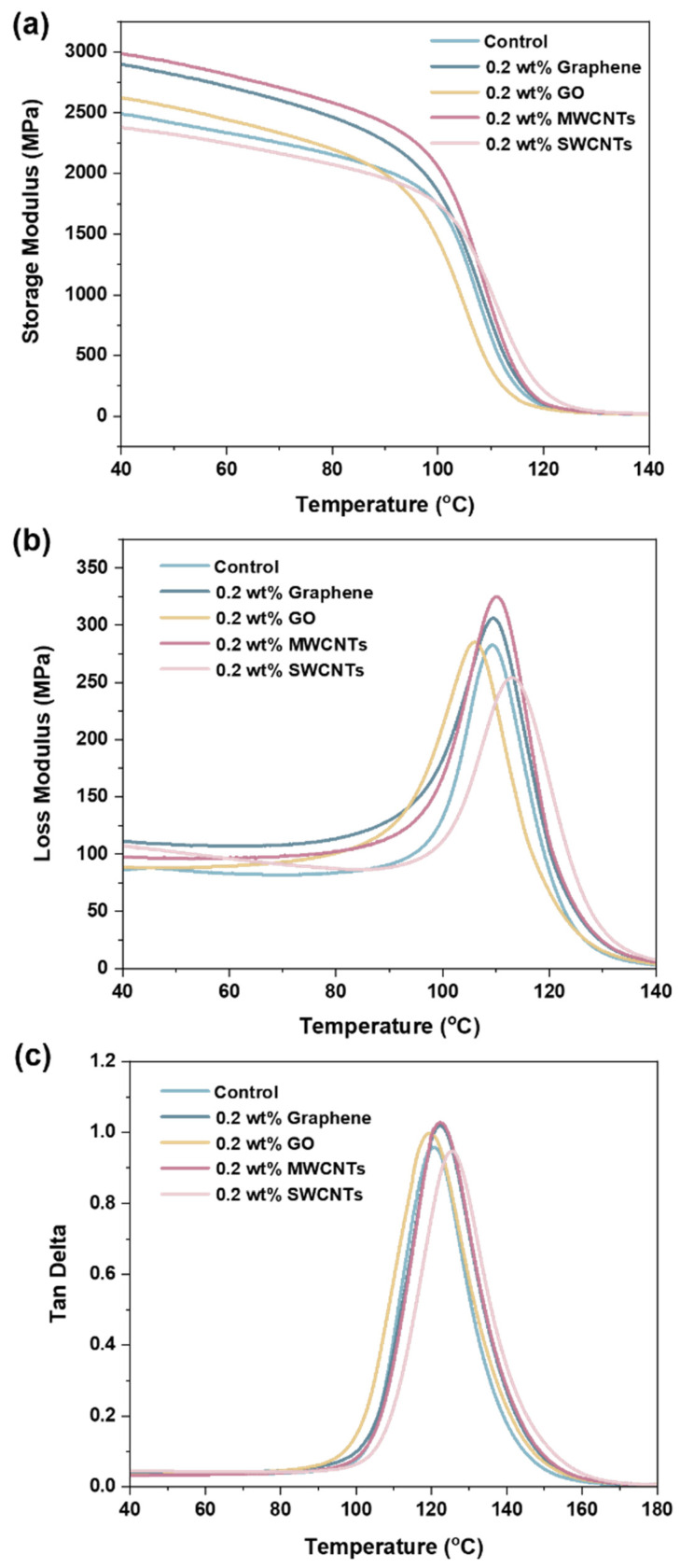
The curves of storage modulus (**a**), loss modulus (**b**), and tan δ (**c**) of the control vinyl ester resin and four composites as a function of temperature.

**Table 1 polymers-15-04536-t001:** Thermo-mechanical properties of the control vinyl ester resin and four composites.

Sample	Storage Modulus (MPa)	Maximum Loss Modulus (MPa)	*T*_g_ (°C)
Control	2490	283	121
Graphene	2898	307	122
GO	2624	285	120
MWCNTs	2986	325	122
SWCNTs	2381	253	125

## Data Availability

Data are contained within the article.

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
