# Peer review of "A Comparison Study on Toughening Vinyl Ester Resins Using Different Nanocarbon Materials"

_polymers, 2023, doi:10.3390/polym15234536_

Round 1

Reviewer 1 Report

Comments and Suggestions for Authors

This work is devoted to investigate the effect of various nanocarbon materials on the strengthening of vinylester resin composites.  In this study, carbon nanotubes (CNTs), graphene nanoplatelets (GNPs) and carbon nanofibers (CNFs) were selected as nanocarbon additives.

1. The abstract and conclusions need to be finalized.

2. In the review, evaluate the methods of synthesis of carbon nanotubes as it affects their properties (Shchegolkov A.V., Shchegolkov A.V. Synthesis of carbon nanotubes using microwave radiation: technology, properties and structure. Russ J Gen Chem 92, 1168-1172 (2022)) and thus the properties of the composite. ttps://link.springer.com/article/10.1134/S1070363222060329.

3. Provide a comparative table summarizing the results of similar studies by other authors.

Comments on the Quality of English Language

 Minor editing of English language required

Author Response

Please find the response in the attached word file.

Reviewer 2 Report

Comments and Suggestions for Authors

Liu et al studied investigate influence of different carbon allotropes for reinforcing vinyl ester resins. This work summarizes valuable insights on use of these carbon based fillers to make the resins useful for tough and stiffer industrial applications. However, some sections need attention and must be revised before final acceptance. Some points are –

[1] Abstract can be improved by summarizing improved values from experiments and avoid talking about characterization techniques like SEM etc. Also, please provide which is the best filler among the carbon allotropes studied in abstract and conclusion clearly?

[2] Introduction section is very weak. At least, one more paragraph must be dedicated to the vinyl ester resins and one more with carbon allotropes discussing them in more details. Finally, the last paragraph must state the novelty of the work such as why this work is important, what are its advancement from existing literature and so on.

[3] The material section is extremely poor. Somewhere authors are using abbreviations, somewhere in full form. They should be uniform. Then, please provide the missing details of these carbon nanomaterials such as their aspect ratio, diameters, surface areas, lateral lengths, shapes, surface functionalities and densities.

[4] In section 2.2, why authors choose 0.2 wt% of fillers? What is the practical significance of this concentration must be justified? Moreover, did authors optimized the processing conditions used for fabricating different composites via different methods? In summary, please also comment on which method is best and which is worst and why? In characterization section, authors should also provide the experimental details of optical tests and SEM tests?

[5] For all unfiled composites, the “Neat” should be “control” or “unfilled” sample. Please change this in all concerned graphs?

[6] In section 3.2, authors talk about the surface functionality and their impact in improving binding energies or filler-polymer interface? Please provide some spectroscopy technique like FTIR or XPS to demonstrate these aspects, If author not perform, please provide a literature support to these claims.

[7] The present optical image or SEM provide only macroscopic view of filler dispersion. The nanoscale view of fillers like CNTs or GO or GNPs is missing. So, please provide the high resolution SEMs, or TEMs or AFMs.

[8] the title of section 3.4 is wrong, these measurements are not called “thermo-mechanical properties” but they are “dynamic mechanical analysis” or “dynamic mechanical thermal analysis”. So, please correct the title.

[9] The conclusion is too general. It should be re-written and must highlights (a) important values and highlights of the experimental output (b) which is best filler among the carbon nanomaterials studied (c) what is the key takeaways from the study (d) why this work is important from existing literature; and (e) what are the future prospects of this work, challenges, advantages and industrial vision?

Good Luck with the revisions!

Comments on the Quality of English Language

Minor editing of English language required. 

Author Response

Please find the response in the attached word file 

Round 2

Reviewer 1 Report

Comments and Suggestions for Authors

Accept in present form

Reviewer 2 Report

Comments and Suggestions for Authors

Paper is acceptable in present form